# A Case for a Maternal Culturally Tailored Smoking Cessation Research Agenda

**DOI:** 10.3390/ijerph21111414

**Published:** 2024-10-25

**Authors:** Danyetta D. Anderson, Tracy R. McKnight

**Affiliations:** Tobacco Related Disease Research Program, Research Grants Program Office, Office of the President, University of California, 11th Floor, 111 Franklin Street, Oakland, CA 94607, USA; tracy.richmond-mcknight@ucop.edu

**Keywords:** AA (African American), BIPOC (Black; Indigenous; and People of Color), AIAN (American Indian/Alaskan Native), MMRC (Maternal Mortality Review Committee), pregnancy-related deaths, maternal mortality, infant mortality, perceived stress, social and political determinants of health, smoking cessation disparities, healthcare disparities, culturally tailored smoking cessation

## Abstract

Background/Objectives: Despite national efforts, smoking rates during pregnancy remain high among certain demographics, particularly American Indian/Alaska Native and younger women. This study examines the causal link between maternal smoking, maternal and fetal mortality, and social determinants of health, highlighting disparities faced by Black, Indigenous, and People of Color (BIPOC) and American Indian/Alaskan Native (AIAN) pregnant persons. Methods: Data from various sources, including national reports and committee findings, were analyzed to assess trends in maternal smoking, mortality rates, and associated factors. While smoking rates among all groups have declined, disparities persist. Young women, BIPOC, and American Indian/Alaska Native women, and those with lower educational attainment, have higher smoking rates. Black women exhibit significantly higher maternal mortality rates, often linked to cardiac/coronary conditions. Stress, exacerbated by social determinants of health like poverty and housing insecurity, emerges as a key factor driving smoking behavior, particularly among African Americans. The leading causes of pregnancy-related deaths vary by race and ethnicity, with preventability noted in 80% of cases. Perinatal exposure to cigarette smoking is also identified as a leading cause of poor infant health outcomes, emphasizing the importance of addressing smoking behavior during and after pregnancy. Results: This report advocates for a comprehensive approach to reducing maternal and fetal mortality rates, with a focus on adapting existing smoking cessation programs to adopt culturally tailored agendas in order to address social and political determinants of health as well as behavioral drivers of tobacco use among pregnant persons.

## 1. Introduction

Despite the success of nationally recognized commercial tobacco cessation programs, the 2018 National Center for Health Statistics data brief substantiates existing reports that smoking rates during pregnancy for American Indian/Alaska Native groups are among the highest of any racial ethnic minority group. This trend in smoking prevalence is also seen among women who earned a high school diploma or general education degree (GED) (12.2%), followed by women who earned less than a high school diploma (11.7%), and women with some college credit or who had earned at least an associate’s degree (7.9%) [1]. The increased risk of maternal smoking among low socioeconomic groups compared to higher SES, and among low income and low educational attainment groups, are also reflected in other developing countries, including Canada [2], Finland [3], Scotland [4], Australia [5], and Iceland [6].

In a retrospective cohort study using the US’s national linked birth/infant death data, 13.2% of pregnant women who delivered live births reported maternal smoking. Compared to nonsmoking pregnant women, infant mortality rates were 40% higher in the group of pregnant women who smoked. A dose-dependent trend also showed an increased risk of infant mortality with the number of cigarettes smoked prenatally [7]. Further increasing the adverse burden of maternal smoking is the risk for sudden unexpected infant death (SUID) more than doubles with any maternal smoking during pregnancy, and if causality is assumed, 22% of SUIDs can be directly attributed to maternal smoking in the US [8]. The inequities in maternal smoking and infant mortality rates highlight the importance of the inclusion of healthcare disparities as well as the social determinants of health as critical variables that should be addressed when considering maternal smoking cessation efforts.

## 2. Interlinked Disparities

The National Center for Health Statistics (NCHS) report maternal mortality rates in 2020 for non-Hispanic Black women as 55.3 deaths per 100,000 live births, representing a 3–4 times higher mortality risk than that of their white counterparts. In 2021, the maternal mortality rates for Black women, which also correlated with higher maternal age, increased to 69.9 per 100,000 [9]. These emerging data on mortality rates within the US were also reflected by a Maternal Mortality Review Committees (MMRCs) report, which found that for non-Hispanic Blacks, the leading underlying cause of death was related to cardiac and coronary conditions (e.g., cardiac and coronary conditions include deaths of coronary artery disease, pulmonary hypertension, acquired and congenital valvular heart disease, vascular aneurysm, hypertensive cardiovascular disease, Marfan Syndrome, conduction defects, vascular malformations, and other cardiovascular disease; and excludes cardiomyopathy and hypertensive disorders of pregnancy), which also mimic maternal complications from smoking. However, the most impactful outcome of report was the finding that 80% of pregnancy-related deaths were determined to be preventable [10].

This disparity in maternal mortality is on trend with the National Vital Statistics Report, which also reflects an increase in mortality rates for infants in each respective ethnic group, with the infants of non-Hispanic Black women having rates more than twice as high as those for infants of non-Hispanic whites [11]. Unraveling the interlinked disparities in maternal and fetal mortality necessitates not only an evaluation of the causal links of smoking during pregnancy and the engenderment to the mother’s health (e.g., risk of cancer, cardiovascular disease, and abnormal bleeding, further exacerbating other comorbidities), but also a consideration of perinatal exposure to the insults of nicotine (e.g., risk of congenital disabilities, sudden infant death syndromes (SIDS), premature birth weight, neurodevelopmental and behavioral problems, obesity, hypertension, type 2 diabetes, impaired lung function, and asthma and wheezing) [12].

## 3. Enduring Challenges

Investigating inequities in maternal tobacco smoking necessitates an examination of the social and political determinants of health that perpetuate negative social and behavioral patterns that adversely affect health outcomes in ethnic minority enclaves. Within the United States, racial and ethnic minorities historically encounter higher rates of chronic disease and premature deaths compared to whites. Incidentally, the “immigrant paradox” which was once deemed to be a protective health outcome has been shown to have diminishing returns the longer one lives within the United States [13]. While there have been inroads in achieving healthy outcomes in minority communities, highlighting the disparities in the disease burden among ethnic minority groups compared to whites is evidence that disparities persist. The National Center for Health Statistics (NCHS) reports that African American women have the highest percentage of preterm singleton births, at 11.1 percent, with Asian or Pacific Islander women having the lowest, at 6.8 percent, among the five racial and ethnic groups measured [14]. For the Indigenous population, the infant mortality rates are much worse, at 60% higher than those of their white counterparts [15]. In consideration of the political determinants of health which Daniel E. Dawes argues are the social drivers that affect all other dynamics of health—including poor environmental conditions, inadequate transportation, unsafe neighborhoods, and lack of health food options—it is important to evaluate the impact of these overarching themes and their integration within the ongoing challenges in addressing healthcare disparities [16]. Historically, low-income communities experience more chronic stress and engage in more negative health behaviors, particularly smoking, to mitigate adverse mental health outcomes and persistent health inequity [17]. Even perceived stress, which is associated with the feeling or idea of how much stress one is experiencing, has been shown in several cross-sectional studies to result in greater odds of self-medicating by smoking [18,19]. Furthermore, alleviating stress by smoking plays a contributory part in engaging in persistent smoking among African Americans [20,21]. In a community of Australian adults, it was shown that depression and anxiety symptoms were not only associated with a higher prevalence of smoking, but compared to those with a tertiary education, the risk of smoking increased among those with a lower level of education as well (12% to 25%). This outcome of the Australian study supports findings from other countries which substantiate the impact of mental health parameters as important contributory factors in increasing smoking prevalence [22,23]. Several studies have further shown a strong positive association between perceived stress and nicotine withdrawal symptomatology in women [24,25]. Environmentally, it is also shown that African Americans are less likely than white smokers to have a total smoking ban in their homes, which means that home smoking policies do not have the same impact on cessation outcomes for African Americans [26]. Furthermore, a staggering 84.5% of African Americans who smoke use menthol-flavored cigarettes, a chemical additive which makes it easier to groom a first-time smoker to cigarettes and makes it harder to quit [27,28].

Collectively, these findings emphasize the importance of better understanding how environmental influences as well as behavioral health can impact chronic and perceived stress within communities with adverse conditions. The investigation of effective maternal tobacco smoking strategies must be shaped by a thoughtful and deliberate research agenda focused on integrating health equity policies and treatment efforts that consider the multidimensional measure of healthcare disparities and its impact on ethnic and minority communities within this country.

## 4. Conclusions

Similar to the establishment of the Maternal Mortality Review Committees (MMRCs) in the US, several other countries have successfully adopted a similar surveillance of pregnancy outcomes to better inform public health agencies. In England, a recent policy change, which resulted in an increased contribution to cessation support in pregnancy provided by The National Health Service (NHS), galvanized both the local authority and NHS trusts in its delivery of smoking cessation services to pregnant women. The recommendations to incorporate the most effective smoking cessation support provided by NHS trusts as well as clinicians were used to both inform and define policy recommendations adopted by the UK government. Furthermore, for any professional involved in disseminating smoking cessation support to pregnant persons, training is performed by a nationally recognized training program to ensure that any support commissioned by various organizations is evidence-based and consistent with the recommendations of the governing body [29]. Addressing the multidimensional measure of structural and political determinants of health is crucial for mitigating disparities in maternal and fetal morbidity and mortality among racial and ethnic minorities. Systemic racism and discrimination permeate healthcare systems throughout the US, further contributing to skepticism among BIPOC, AIAN, and African American communities regarding participation in research studies [30]. Treatment protocols often fail to account for cultural nuances, language preferences, and traditional healing practices, leading to cultural insensitivity that alienates potential participants. To bridge this gap, researchers must actively engage with community leaders and elders and ensure that interventions are crafted with cultural sensitivity and respect for the customs within each community. Furthermore, addressing smoking-related adverse pregnancy outcomes within vulnerable communities requires that existing smoking cessation programs are adaptive and inclusive of social and political determinants of health. Data collection that is informed by the conditions in which people are born, grow, work, live, and age, must be considered when evaluating a political, social, and research agenda that informs efforts in addressing maternal and fetal health outcomes. Furthermore, the need to develop culturally tailored smoking cessation programs that incorporate strategies to address factors contributing to chronic and perceived stress should be explored as part of sustainable interventions tailored to subpopulations with disparate maternal and fetal mortality rates. In conclusion, understanding the intersectionality of social factors and biology in research dedicated to developing, implementing, and evaluating commercial tobacco cessation interventions is vital for addressing health outcomes among some of the most vulnerable members of our population: pregnant individuals who identify as Black, Indigenous, and People of Color (BIPOC), American Indians, and Alaska Native (AIAN).

## Data Availability

No new data was created in this manuscript preparation.

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
