# Peer review of "A Case for a Maternal Culturally Tailored Smoking Cessation Research Agenda"

_ijerph, 2024, doi:10.3390/ijerph21111414_

Round 1

Reviewer 1 Report

Comments and Suggestions for Authors

This brief report aims to present a concise rationale for a maternal culturally tailored smoking cessation research agenda. The authors provide a review of the current literature, discussing smoking prevalence and disparities in smoking rates; maternal health disparities; particularly for non-Hispanic Black women, and the potential role of social determinants of health on maternal mortality.  This is an important topic and generally the review of the literature is clear and compelling in some places. However, in it’s current form this brief report seems to fall short of it’s aim.  Specifically, the following questions are not directly addressed: how does maternal smoking related to maternal morbidity and mortality?  Is maternal smoking an important driver of maternal morbidity and mortality? Why are culturally tailored interventions needed? For the latter question, the conclusion section does begin to address this but it could be further discussed with relevant citations.

Specific comments

Introduction

Line 36 – it might be helpful to know what the smoking rates were for women under age 30 and non-Hispanic American Indian or Alaska Native and non-Hispanic White women.

Line 39 - PRAMS monitors 46 sites currently https://www.cdc.gov/prams/php/participating-states/index.html, however, the author correctly sites the Tong et al, 2013 report. Perhaps phrasing this to represent the paper discussed rather than a description of PRAMS more generally would be more appropriate.

Line 51 – The section on interlinked disparities does not articulate the association with smoking rates.

Line 66 – the section Limited Data, Enduring Challenges this section briefly mentions social determinants of health, connects to stress and smoking, and factors impacting African American people who smoke. However, it does not provide a cohesive explanation of what the data gaps are or what the on-going challenges are.

Line 67- This first sentence is missing a subject – maybe “people experiencing poor living wages?”.

Line 74-75 – The precise meaning of the statement “poor mental health outcomes are exaggerated or initiated” is unclear.

The conclusion provides several clear statements regarding gaps in the literature, challenges to conducting high-quality research and the importance of attention to culture.

Line 99 – This statement “the obstacles to addressing health inequity…. are imperative and critical to informing data collection” is unclear. This should be re-worded to more accurately describe how data collection may be used in addressing obstacles.  

Comments on the Quality of English Language

There are a few places where your sentences may be missing words or the meaning is not clear. 

Author Response

However, in its current form this brief report seems to fall short of its aim.  Specifically, the following questions are not directly addressed: how does maternal smoking related to maternal morbidity and mortality?   Is maternal smoking an important driver of maternal morbidity and mortality?  Why are culturally tailored interventions needed? For the latter question, the conclusion section does begin to address this but it could be further discussed with relevant citations.

Response: As it relates to the first two questions, this brief report is not intended to directly address how maternal smoking is related to maternal morbidity and mortality.  Inherit to the manuscript title is the implied intent of the report which is to present a case for further investigation and perhaps a deeper discussion of the causality that maternal smoking may have on the wider debate of maternal and fetal morbidity and mortality in communities of color.   

Furthermore, the report seeks to make the case for culturally tailored smoking interventions in minority communities by highlighting the impact of social and political determinants of health, which are rooted in structural racism and are not adequately addressed in nationally recognized smoking cessation programs.  Hopefully, these ideas are better articulated following the revisions to the manuscript.    

Specific comments

Introduction

Line 36 – it might be helpful to know what the smoking rates were for women under age 30 and non-Hispanic American Indian or Alaska Native and non-Hispanic White women. Addressed within introduction

Line 39 - PRAMS monitors 46 sites currently https://www.cdc.gov/prams/php/participating-states/index.html, however, the author correctly sites the Tong et al, 2013 report. Perhaps phrasing this to represent the paper discussed rather than a description of PRAMS more generally would be more appropriate.  Agree, not relevant.  Idea was reframed within introduction.

Line 51 – The section on interlinked disparities does not articulate the association with smoking rates. Addressed on page 2; lines 71-81

Line 66 – the section Limited Data, Enduring Challenges this section briefly mentions social determinants of health, connects to stress and smoking, and factors impacting African American people who smoke. However, it does not provide a cohesive explanation of what the data gaps are or what the on-going challenges are.  Section renamed: Enduring Challenges; Addressed on page 2 and page 3

Line 67- This first sentence is missing a subject – maybe “people experiencing poor living wages?”.  Sentence and idea restated in the section titled “Enduring Challenges”

Line 74-75 – The precise meaning of the statement “poor mental health outcomes are exaggerated or initiated” is unclear.  Idea restated in the section titled “Enduring Challenges”  

The conclusion provides several clear statements regarding gaps in the literature, challenges to conducting high-quality research and the importance of attention to culture.  Thank you

Line 99 – This statement “the obstacles to addressing health inequity…. are imperative and critical to informing data collection” is unclear. This should be re-worded to more accurately describe how data collection may be used in addressing obstacles.  Idea restated and addressed in the conclusion.   

Comments on the Quality of English Language

There are a few places where your sentences may be missing words or the meaning is not clear. Thank you, grammar and syntax were addressed throughout the manuscript.

Reviewer 2 Report

Comments and Suggestions for Authors

Dear Authors, your text seems like a heartfelt appeal to the institutions. It is written correctly and with the right bibliographical references. It deserves publication, hoping that politicians can draw inspiration from your article.

Please please make the way of numbering bibliographic entries homogeneous. (see lines 72-76 page 2)

Author Response

Dear Authors,

Your text seems like a heartfelt appeal to the institutions. It is written correctly and with the right bibliographical references. It deserves publication, hoping that politicians can draw inspiration from your article.

Thank you for your feedback on this manuscript.  It is also my hope that politicians can draw inspiration from the recommendations in this article. 

Please make the way of numbering bibliographic entries homogeneous. (see lines 72-76 page 2). 

Thank you for this observation.  The bibliographic entries have been updated to be consistent throughout the manuscript as well as in the final bibliography list. 

Reviewer 3 Report

Comments and Suggestions for Authors

The brief report is very well written. I have two minor suggestions that may improve the quality of the paper:

1. Page 2, Line 45: "in US populations with the highest maternal smoking prevalence"  It would be helpful if the authors could list the specific prevalence for comparison.

2. It would be ideal if the authors could compare the U.S. with other countries. Specifically, how have other countries addressed potential disparities in maternal and fetal mortality rates related to smoking? Are there any programs or successful interventions that the U.S. could learn from and implement?

Author Response

  1. Page 2, Line 45: "in US populations with the highest maternal smoking prevalence"  It would be helpful if the authors could list the specific prevalence for comparison. Yes, this prevalence is better articulated in the updated text. (Page 1, line 36-40)  
  2. It would be ideal if the authors could compare the U.S. with other countries. Specifically, how have other countries addressed potential disparities in maternal and fetal mortality rates related to smoking? Are there any programs or successful interventions that the U.S. could learn from and implement?  Yes, where possible, I have included comparisons of the U.S. with other countries as appropriate. (Page 1, line 42; Page 3 line 111; 114)  Unfortunately, this brief report cannot accommodate the entire span of programs or successful interventions that the U.S. could learn from and implement; however, I made reference to other countries adoption of pregnancy surveillance as well as highlighting a successful smoking cessation approach adopted in the UK. (Page 4, lines 133-145)  

Round 2

Reviewer 1 Report

Comments and Suggestions for Authors

Overall, the manuscript is substantially strengthened as a result of the revisions. 

One minor comment:

This sentence is somewhat unclear - the trend described in the previous paragraph shows increasing maternal mortality:  "This disparity in maternal mortality is on trend with the National Vital Statistics Report that showed a decline in mortality rate for infants in each respective ethnic groups,  with non-Hispanic black women having rates more than twice as high for infants of non- Hispanic whites. 18"

Author Response

Second round revisions: Reviewer #2:

One minor comment:

This sentence is somewhat unclear - the trend described in the previous paragraph shows increasing maternal mortality:  "This disparity in maternal mortality is on trend with the National Vital Statistics Report that showed a decline in mortality rate for infants in each respective ethnic groups,  with non-Hispanic black women having rates more than twice as high for infants of non- Hispanic whites. 12"

Thank you for your comment.  Yes, the word “decline” in that sentence, line 73 page 2, should have read “increase.” The change has been updated within the text to reflect the trends in both maternal and fetal mortality rates for each respective minority groups as reported in both National reports.